# Task-Agnostic Amortized Inference of Gaussian Process Hyperparameters

**Sulin Liu, Xingyuan Sun, Peter J. Ramadge, Ryan P. Adams**

Princeton University
{sulinl, xs5, ramadge, rpa}@princeton.edu

## Abstract

Gaussian processes (GPs) are flexible priors for modeling functions. However, their success depends on the kernel accurately reflecting the properties of the data. One of the appeals of the GP framework is that the marginal likelihood of the kernel hyperparameters is often available in closed form, enabling optimization and sampling procedures to fit these hyperparameters to data. Unfortunately, point-wise evaluation of the marginal likelihood is expensive due to the need to solve a linear system; searching or sampling the space of hyperparameters thus often dominates the practical cost of using GPs. We introduce an approach to the identification of kernel hyperparameters in GP regression and related problems that sidesteps the need for costly marginal likelihoods. Our strategy is to "amortize" inference over hyperparameters by training a single neural network, which consumes a set of regression data and produces an estimate of the kernel function, useful across different tasks. To accommodate the varying dimension and cardinality of different regression problems, we use a hierarchical self-attention-based neural network that produces estimates of the hyperparameters which are invariant to the order of the input data points and data dimensions. We show that a *single* neural model trained on synthetic data is able to generalize directly to several *different unseen* real-world GP use cases. Our experiments demonstrate that the estimated hyperparameters are comparable in quality to those from the conventional model selection procedures, while being much faster to obtain, significantly accelerating GP regression and its related applications such as Bayesian optimization and Bayesian quadrature. The code and pre-trained model are available at `https://github.com/PrincetonLIPS/AHGP`.

## 1 Introduction

Gaussian processes (GPs) are powerful tools for modeling distributions over functions. They are highly flexible Bayesian nonparametric models for which the posterior is often available in closed form. GPs are successful models for a variety of machine learning tasks, from regression and classification [53], to Bayesian optimization [65], to modeling of dynamics [36, 70]. The predictive performance of a Gaussian process, however, is highly dependent on the specifics of the prior on functions, as determined by the associated positive definite kernel function [53]. To find a good prior, one needs to first come up with a family of kernel functions that is capable of capturing the structure of the data.

The kernel hyperparameters must then be determined from data, usually by maximizing the log marginal likelihood (MLL), i.e., empirical Bayes [53]. There are two major issues associated with this procedure. First, evaluating the log marginal likelihood (and its gradient) for different hyperparameters is generally expensive, with $\mathcal{O}(N^3)$ computational complexity for $N$ training data. Second, the log MLL is usually highly non-concave, making both sampling and optimization difficult.

Getting stuck in bad local maxima or saddle points can result in hyperparameters with significant model mismatch and thus poor predictive performance [51, 45, 77]. Alternatively, to avoid the catastrophic failure of bad local minima and point-estimates in MLL optimization, one may attempt a full Bayesian treatment by integrating out the kernel hyperparameters using Markov chain Monte Carlo (MCMC) techniques [43, 16, 17]. In this paper, we focus on the optimization case, although many of the ideas we present could also be applied to marginalization of hyperparameters.

Various approaches to scaling Gaussian processes have been proposed, exploring innovative ideas such as intelligently selecting a subset of the data [63, 72] or constructing a low-rank approximation of the covariance matrix based on virtual "inducing" points [8, 50, 60, 64, 67, 28]. These scaling approaches have generally focused on how to solve the linear system more quickly without having to significantly compromise the model or the predictive performance, either by reducing the size of the linear system, solving the linear system approximately, or approximating the model with one that has computationally convenient structure. Focusing on the linear system has been a sensible approach, as effective solutions simultaneously enable both direct prediction with the Gaussian process and evaluation of hyperparameters via the log marginal likelihood.

Here however, we take an entirely different approach and focus solely on the model selection problem, without reference to linear systems at all. Instead, we *amortize* the Gaussian process model selection problem by training a neural network to consume input/output observations and emit an estimate of the hyperparameters that would otherwise arise from maximizing the log marginal likelihood. This approach is inspired by amortized variational inference approaches [33, 55, 26, 56], which similarly sidestep expensive optimization procedures in favor of directly producing estimates. In the variational inference case, the neural network produces approximate posterior distributions over the unknown latent variables; here we produce point estimates of the unknown hyperparameters.

Of course, as noted above, selection of the kernel family is just as important as determination of hyperparameters, and we view this as a crucial piece of the amortized model selection puzzle. One approach to modeling the huge space of valid kernel functions is to represent the target kernel as a composition of different base kernels [12, 39, 66]. In principle, various base kernels, composition rules, and associated hyperparameters could be modeled as latent random variables, and emitted by a neural network. However, since the latent variables here involve a mix of discrete variables (types and combinations of kernel functions) and continuous variables (hyperparameters associated), we have found this approach to be difficult. Moreover, the interrelated effects of, e.g., length scale and kernel choice, make it difficult to unify the hyperparameter space across composed kernels. Thus, instead of working with compositions of commonly-used kernels, we focus on stationary kernels in the spectral domain and directly learn the spectral density of the kernel function [73, 57, 54]. The space of spectral densities provides a unified and compact continuous representation of the space of stationary covariance functions. In particular, we model the spectral density of the kernel function as a mixture [73] that is capable of approximating any stationary kernel arbitrarily well.

There are two particularly salient challenges associated with training a single neural network to produce effective hyperparameters for many different regression-type tasks: both the amount of data and the dimensionality of the input can vary from problem to problem. To address this, we develop a specialized hierarchical self-attention structure that consumes datasets and produces spectral densities while being invariant to permutations of the data and the dimensions. Thus, this single "meta-model" can be applied to different problems for which a Gaussian process is applicable. The parameters of the network are trained using gradients computed via reverse-mode automatic differentiation through the log marginal likelihood of the Gaussian process, for randomly-generated synthetic data from the prior. Then we directly apply the trained neural model to real-world datasets of varying size and dimension in different GP applications. Even though the model is trained with only synthetic data, experimental evidence indicates that the estimated hyperparameters are comparable in quality to those from the conventional model selection procedures while being $\sim 100$ times faster to obtain.

## 2   Gaussian Processes

In this section, we establish background concepts and notations necessary for the discussion of the amortized hyperparameters inference approach described in Section 3.

A Gaussian process defines a distribution over functions $f : \mathcal{X} \to \mathbb{R}$, and is specified by its mean function $\mu(\mathbf{x})$ and positive-definite covariance function $k(\mathbf{x}, \mathbf{x}')$, where $\mathbf{x}, \mathbf{x}' \in \mathcal{X}$:

$$f(\mathbf{x}) \sim \mathcal{GP}\left(\mu(\cdot), k\left(\cdot, \cdot\right)\right), \qquad \mu(\mathbf{x}) = \mathbb{E}[f(\mathbf{x})], \qquad k\left(\mathbf{x}, \mathbf{x}'\right) = \text{cov}\left(f(\mathbf{x}), f(\mathbf{x}')\right) \quad (1)$$

For any finite set of points in $\mathcal{X}$, $\mathbf{X} := \{\mathbf{x}_1, \ldots, \mathbf{x}_N\}$, the corresponding function values $\mathbf{f} := (f(\mathbf{x}_1), \cdots, f(\mathbf{x}_N))^\mathsf{T}$ follow a multivariate Gaussian distribution: $\mathbf{f} \sim \mathcal{N}(\boldsymbol{\mu}, \mathbf{K}_{\mathbf{XX}})$, where $[\boldsymbol{\mu}]_i = \mu(\mathbf{x}_i)$ and $[\mathbf{K}_{\mathbf{XX}}]_{ij} = k(\mathbf{x}_i, \mathbf{x}_j)$. For a training dataset $\mathcal{D} = \{(\mathbf{x}_i, y_i)\}_{i=1}^N$, each $y_i$ is commonly assumed to be generated by adding an i.i.d. zero-mean Gaussian noise to $f(\mathbf{x}_i)$, i.e., $y_i = f(\mathbf{x}_i) + \epsilon_i$, where $\epsilon_i \sim \mathcal{N}(0, \sigma_\epsilon^2)$. Denote $\mathbf{y} := [y_1, \cdots, y_N]^\top \in \mathbb{R}^{N \times 1}$. For new data input $\widetilde{\mathbf{X}} := \{\widetilde{\mathbf{x}}_1, \ldots, \widetilde{\mathbf{x}}_{N'}\}$ of size $N'$, the Gaussianity of the prior and likelihoods make it possible to compute the predictive distribution in closed form:

$$\widetilde{\mathbf{f}} | \widetilde{\mathbf{X}}, \mathcal{D} \sim \mathcal{N}\left(\widetilde{\boldsymbol{\mu}}, \mathbf{K}_{\widetilde{\mathbf{f}}}\right), \quad \widetilde{\boldsymbol{\mu}} = \mathbf{K}_{\widetilde{\mathbf{X}}\mathbf{X}}(\mathbf{K}_{\mathbf{XX}} + \sigma_\epsilon^2 \mathbf{I})^{-1}\mathbf{y}, \quad \mathbf{K}_{\widetilde{\mathbf{f}}} = \mathbf{K}_{\widetilde{\mathbf{X}}\widetilde{\mathbf{X}}} - \mathbf{K}_{\widetilde{\mathbf{X}}\mathbf{X}}(\mathbf{K}_{\mathbf{XX}} + \sigma_\epsilon^2 \mathbf{I})^{-1}\mathbf{K}_{\mathbf{X}\widetilde{\mathbf{X}}},$$

where $\mathbf{K}_{\mathbf{X}\widetilde{\mathbf{X}}} \in \mathbb{R}^{N \times N'}$ with $[\mathbf{K}_{\mathbf{X}\widetilde{\mathbf{X}}}]_{ij} = k(\mathbf{x}_i, \widetilde{\mathbf{x}}_j)$.

**Choice of kernel function.** The choice of kernel function is crucial to Gaussian process generalization, as different kernel functions impose various model assumptions, e.g., smoothness, periodicity, etc. (See Chapter 4 of Rasmussen and Williams [53] for an extensive discussion.) If the problem has a known structure, one can sometimes choose a kernel to capture it. Otherwise, kernel learning must be performed by defining an expressive space of kernel functions and selecting the best one through optimization [73, 75, 66] or search [12, 39].

**Hyperparameter inference.** Beyond the particular choice of kernel, it is also common for the covariance function to have so-called *hyperparameters* $\theta$ that govern its specific structure, and the parameterized kernel function is written as $k_\theta(\cdot, \cdot)$. Although a fully-Bayesian treatment is possible [45, 43, 16, 44], the most common approach to determining hyperparameters is to use empirical Bayes and maximize the log marginal likelihood (evidence) with respect to the hyperparameter $\theta$, i.e., perform type II maximum likelihood [3, 41]. The log MLL for observed data $\{\mathbf{X}, \mathbf{y}\}$ is given by:

$$\log p(\mathbf{y}|\mathbf{X}, \theta) = -\frac{1}{2}\mathbf{y}^\top \left(\mathbf{K}_{\mathbf{XX}}(\theta) + \sigma_\epsilon^2 \mathbf{I}\right)^{-1}\mathbf{y} - \frac{1}{2}\log\left|\mathbf{K}_{\mathbf{XX}}(\theta) + \sigma_\epsilon^2 \mathbf{I}\right| - \frac{N}{2}\log 2\pi, \quad (2)$$

where we write $\mathbf{K}_{\mathbf{XX}}(\theta)$ to indicate the dependence of the Gram matrix on the hyperparameters.

To solve the above optimization problem, quasi-Newton methods such as L-BFGS [38] or nonlinear conjugate gradient [30, 19] are usually used. These iterative optimization methods involve taking the gradient of the objective several times for each optimization step. As the gradient of Eqn 2 scales as $\mathcal{O}(N^3)$, this optimization becomes prohibitively expensive on large-scale problems, dominating the computational cost of using GP. Moreover, the non-concavity of the objective in Eqn 2 makes it difficult to ensure convergence to a good maximum.

To address the scaling issue, a low-rank approximation to the kernel matrix is often used either by subsampling the data or via virtual "inducing" points [63, 72, 8, 50, 60, 64, 67, 28, 6, 61]. In general, these methods require inversion of a smaller matrix and reduce the computational complexity to $\mathcal{O}(NM^2)$ ($M$ is the number of subsampled data or "inducing" points), at the cost of a larger and often more challenging optimization problem alongside the potential loss of important information from the dataset. For the special case of the exponentiated quadratic kernel, Burt et al. [5] showed that only $\mathcal{O}(N\log^{2D}(N))$ computational complexity is needed to achieve an arbitrarily good approximation with high probability for input with either compact support or Gaussian distribution of $D$ dimensions.

## 3  Amortized GP Hyperparameter Inference

In this section, we frame the problem of amortized Gaussian process hyperparameter inference. The objective is to avoid the computational overhead and fragility of maximizing the log marginal likelihood, particularly within the inner loop of an iterative procedure such as Bayesian optimization or Bayesian quadrature. Our approach is to instead train a flexible function approximator, i.e., a neural network, to provide high-quality estimates of kernel hyperparameters, directly from the data. We call our method *amortized hyperparameter inference for Gaussian processes* (AHGP).

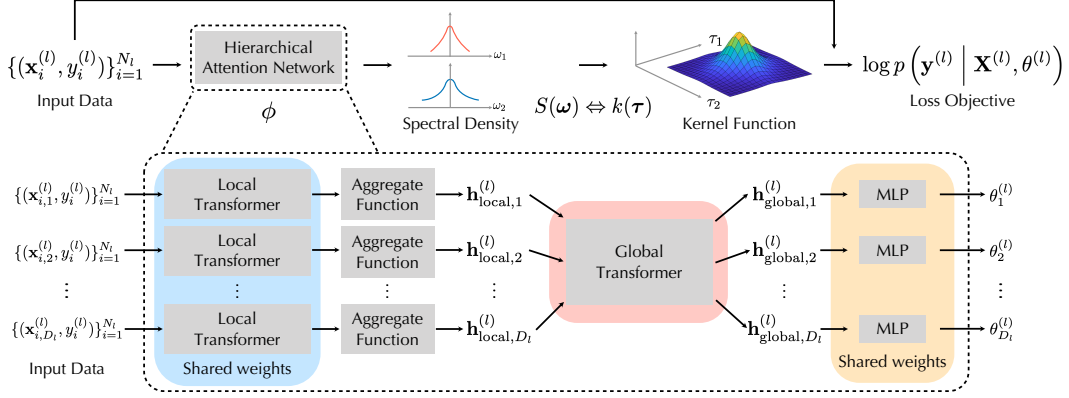

Figure 1: The top part of the figure gives an illustration of the computation graph in AHGP. The bottom part describes our hierarchical attention neural net architecture.

In GP regression, it is common to use stationary kernels, which include most generic covariance functions, such as the exponentiated quadratic, rational quadratic and Matérn. Consistent with this, we will study stationary kernels, but seek to make them highly flexible. One way to achieve flexibility is via composition of different commonly-used base kernels as in Duvenaud et al. [12], Lloyd et al. [39], Sun et al. [66]. Model selection in this approach—choice of base kernels, composition rules, and kernel hyperparameters—could in principle be done via a neural network; however, we have found the associated optimization problem to be difficult and complicated. First, it involves optimization over both discrete (kernel types and composition rules) variables and continuous (hyperparameters) variables. Second, different kernels bring with them continuous parameters with different cardinalities and interpretations, making it challenging to identify a unified hyperparameter space.

## 3.1 Spectral Modeling of Stationary Kernel Functions

In lieu of a compositional approach to the kernel function, we build a flexible approach around the duality between stationary kernels and their spectral density, taking advantage of the well-known theorem by Bochner stated below.

**Theorem 1.** *(Bochner [4]) A complex-valued function $k$ on $\mathbb{R}^d$ is the covariance function of a weakly stationary (also known as covariance-stationary) mean square continuous complex valued random process on $\mathbb{R}^d$ if and only if it can be represented as*

$$k(\tau) = \int_{\mathbb{R}^d} e^{2\pi i \omega^\top \tau} d\mu(\omega), \tag{3}$$

*where $\mu$ is a positive finite measure, often known as the spectral measure of the random process.*

When $\mu$ is absolutely continuous with respect to the Lebesgue measure, its Radon–Nikodym derivative (density) $S(\omega)$ is called the *spectral density* of the random process. The kernel function $k(\cdot)$ and the spectral density $S(\cdot)$ can be understood to be Fourier transform pairs. We take advantage of this correspondence and use a neural network to predict the spectral density of the kernel function rather than the covariance function itself.

Following previous work [73, 74], we model the spectral density via a Gaussian mixture, leading to interpretability and closed form evaluation of the kernel. Additionally, the fact that Gaussian mixtures are dense in the space of probability distribution functions [62, 73] makes them capable of approximating the spectral density of any stationary kernel function arbitrarily well. Here we further assume that the kernel function has a product structure over different dimensions and every dimension has its own mixture of Gaussians. The product kernel structure is a common modeling choice in Gaussian process regression, and arises naturally in many generic kernels such as the exponentiated quadratic and its automatic relevance determination (ARD) version. Additionally, it is common to compose kernels via element-wise products, with each dimension's functional properties encoded in its corresponding kernel function [53, 23, 12, 74].

## 3.2 Formulation

We now formalize this problem of amortized hyperparameter inference in a Gaussian process. We are interested in fitting many different regression functions of the form $f : \mathbb{R}^D \to \mathbb{R}$, with varying values of $D$. For the $l$-th regression task $\mathcal{T}^{(l)}$, a set of input/output training data are observed and are given by $\mathcal{D}^{(l)} := \{(\mathbf{x}_i^{(l)}, y_i^{(l)})\}_{i=1}^{N_l} = \{\mathbf{X}^{(l)}, \mathbf{y}^{(l)}\}$, where $y_i^{(l)} = f_l(\mathbf{x}_i^{(l)}) + \epsilon_i^{(l)}$ with $\mathbf{x}_i^{(l)} \in \mathbb{R}^{D_l}$ and $\epsilon_i^{(l)}$ being i.i.d. zero-mean Gaussian noise. Assume we are given $L$ tasks, with each task randomly sampled i.i.d. from a distribution over tasks, i.e., $\{\mathcal{T}^{(l)}\}_{l=1}^{L} \overset{i.i.d.}{\sim} p(\mathcal{T})$. We further assume that for each task, the function values are generated by some underlying Gaussian process with its own unique kernel hyperparameters. The spectral density of each task's GP is modeled as a mixture of $M$ Gaussian over each dimension as discussed in Sec. 3.1, with weights, means and variances denoted as $\theta_d^{(l)} = \{\{w_{d,m}^{(l)}\}_{m=1}^{M}, \{\mu_{d,m}^{(l)}\}_{m=1}^{M}, \{\sigma_{d,m}^{2\,(l)}\}_{m=1}^{M}\}$ for the $d$-th dimension in task $l$. For compactness, we use $\theta^{(l)} = \{\theta_d^{(l)}\}_{d=1}^{D_l}$ to denote the collective hyperparameters of the spectral density for task $l$.

The neural network, parameterized by $\phi$, defines a function $g_\phi$ from a dataset $\mathcal{D}^{(l)}$ to an estimate of its spectral density hyperparameters $\theta^{(l)}$, i.e., $\theta^{(l)} = g_\phi(\mathcal{D}^{(l)})$. Through the duality of spectral densities and stationary kernel functions, the spectral mixture product (SMP) kernel [74] is given by:

$$k_{\mathrm{SMP}_\theta}(\boldsymbol{\tau}) = \prod_{d=1}^{D} k_{\mathrm{SMP}_{\theta_d}}(\boldsymbol{\tau}_d) \quad \text{with} \quad k_{\mathrm{SMP}_{\theta_d}}(\boldsymbol{\tau}_d) = \sum_{m=1}^{M} w_{d,m} \exp\left\{-2\pi^2 \boldsymbol{\tau}_d^2 \sigma_{d,m}^2\right\} \cos\left(2\pi \boldsymbol{\tau}_d \mu_{d,m}\right),$$

where $\boldsymbol{\tau}_d$ is the $d$-th component of $\boldsymbol{\tau} = \mathbf{x} - \mathbf{x}' \in \mathbb{R}^D$, i.e., the difference of two data points.

We can now train the neural network to produce hyperparameters using a "dataset of datasets", $\{\mathcal{D}^{(l)}\}_{l=1}^{L}$. With the closed form kernel function specified by its spectral density hyperparameters, the averaged negative log marginal likelihood evaluated from Eqn (2) is used as our training objective:

$$\mathcal{L}\left(\phi, \left\{\mathcal{D}^{(l)}\right\}_{l=1}^{L}\right) = -\frac{1}{L}\sum_{l=1}^{L}\frac{1}{N_l}\log p\left(\mathbf{y}^{(l)} \mid \mathbf{X}^{(l)}, \theta^{(l)}\right), \tag{4}$$

where $\theta^{(l)} = g_\phi(\mathcal{D}^{(l)})$ and $N_l$ is the number of data points in the $l$-th dataset. Once the neural network is trained, it can be used to estimate the GP kernel function that would be appropriate for a new set of input/output data $\mathcal{D}^{\mathrm{test}}$ by simply doing a forward pass of the neural model.

## 4 Hierarchical Attention Network for GP Hyperparameter Learning

As described in the previous section, the neural network learns a function from a dataset $\mathcal{D}^{(l)}$ to spectral density parameters $\{\theta_d^{(l)}\}_{d=1}^{D_l}$, determining the GP kernel function for task $l$. As in other deep learning problems, the architecture of the neural network is critical; in particular, the structure of the network must take advantage of available symmetries. In our case, for general purpose inference of GP kernel hyperparameters, we require an architecture that is versatile enough to accommodate datasets of varying input dimension and with different number of data points. Furthermore, the model should be invariant to permutation of both the data and input dimensions. In other words, for a given dataset $\mathcal{D}^{(l)}$, neither shuffling the order of the (exchangeable) data nor shuffling the order of the dimensions should change the resulting estimate of the kernel function. Importantly, the regularization and parameter sharing induced by enforcement of such invariances should enable the neural network to learn better and faster, analogously to convolutional neural networks for images.

**Architecture.** We draw inspiration from multi-head self-attention mechanisms and propose a hierarchical Transformer [68] type of neural network architecture for tackling the problem of learning GP hyperparameters. A general Transformer model has multiple layers and each layer consists of a multi-head self-attention sub-layer followed by a feed forward network with residual connections and layer normalizations. It serves as an autoregressive encoder that maps a set of input data to a set of output representations. In particular, the self-attention sub-layers allow each input datum to attend to the representations of other data and produce context-aware representations. Multiple layers of self-attention enable modeling of high-order non-linear interactions between input representations. For details about multi-head self-attention mechanisms, we refer readers to Appendix A.

Briefly, our network architecture mainly consists of two hierarchically nested Transformer-like blocks. A graphic illustration of the proposed architecture is presented in Fig. 1.

*Local Transformer*: The first transformer block, LocalTransformer, serves as an encoder of the per-dimension local information about the observed function, e.g., length scale, smoothness, periodicity, etc. It takes in a set of input/output data specific to the $d$-th dimension, e.g., $\mathcal{D}_d^{(l)} = \{(\mathbf{x}_{i,d}^{(l)}, y_i^{(l)})\}_{i=1}^{N_l}$, and outputs a corresponding set of representations $\{\mathbf{h}_{i,d}^{(l)}\}_{i=1}^{N_l}$. In LocalTransformer, only interactions within the $d$-th dimension are involved, hence each $\mathbf{h}_{i,d}^{(l)}$ is a context-aware representation of local information about the underlying function around dimension $d$ of datum $(\mathbf{x}_i^{(l)}, y_i^{(l)})$.

*Aggregate Function*: The outputs from LocalTransformer $\{\mathbf{h}_{i,d}^{(l)}\}_{i=1}^{N_l}$ are aggregated through an AggregateFunction that assembles a single local dimension-specific representation $\mathbf{h}_{\text{local},d}^{(l)}$ for the $d$-th dimension. These feature representations $\{\mathbf{h}_{\text{local},d}^{(l)}\}_{d=1}^{D_l}$ provide a summary of each dimension's local information regarding the observed function.

*Global Transformer*: After the dimension-specific local representations $\{\mathbf{h}_{\text{local},d}^{(l)}\}_{d=1}^{D_l}$ are computed, they are fed into a second dimension-level Transformer block, GlobalTransformer, where non-linear interactions between the dimensions are modeled through multiple layers of multi-head self-attention. The final per-dimensional representations $\{\mathbf{h}_{\text{global},d}^{(l)}\}_{d=1}^{D_l}$, which serve as context-aware representations at a global (dimension) level, are further passed through a multi-layer perceptron (MLP) to produce the final spectral density hyperparameters $\{\theta_d^{(l)}\}_{d=1}^{D_l}$.

**Versatility and permutation invariance.** Self-attention enables the model to consume a set of input/output data with arbitrary data cardinality and dimensionality. The versatility of the model makes it general-purpose: one could train a single neural model to predict GP kernel hyperparameters of different tasks with varying data cardinality and dimensionality, as long as the inputs are real-valued. The proposed model also possesses the following permutation equivariance/invariance properties.

**Proposition 1.** *If* AggregateFunction *is permutation invariant, and weights of* LocalTransformer *and* MLP *are shared across dimensions, then the proposed neural network is permutation equivariant with respect to data dimensions and permutation invariant with respect to data points.*

Intuitively, these properties are inherited from the permutation equivariance of self-attention and the permutation invariance of the AggregateFunction. See Appendix B for a detailed proof. It is also noted that the versatility and inductive biases (permutation invariance/equivariance) of the proposed hierarchical attention network are generic, making our proposed neural model potentially useful for other tasks that involve learning representations over datasets, such as learning statistics of datasets in Edwards and Storkey [13].

**Complexity analysis.** For each multi-head self-attention layer, the computational complexity is $\mathcal{O}(h \cdot n^2)$, where $h$ is the representation dimension and $n$ is the size of the input set. Assuming that LocalTransformer has $l_1$ layers with representation dimension $h_1$ and GlobalTransformer has $l_2$ layers with representation dimension $h_2$, the complexity of our model is $\mathcal{O}(l_1 \cdot h_1 \cdot N^2 + l_2 \cdot h_2 \cdot D^2)$, where $N$ is the number of data points and $D$ is the dimensionality. In comparison, exact marginal likelihood optimization scales as $\mathcal{O}(r \cdot N^3)$, where $r$ denotes the number of gradient evaluations during optimization. It is possible to further reduce the complexity of our model to $\mathcal{O}(l_1 \cdot h_1 \cdot m \cdot N + l_2 \cdot h_2 \cdot m \cdot D)$ if we restrict the number of attentions to $m$ by either introducing sparse attentions [7] or inducing points [37], which we leave for future work.

## 5   Experimental Results

To empirically evaluate the AHGP, we studied three different GP use cases: regression, Bayesian optimization and Bayesian quadrature.

**Baselines.** We compare our method to the standard approach of maximizing the log marginal likelihood with respect to hyperparameters. We also compare with the sparse variational Gaussian processes method (SGPR) [67, 28], which uses inducing points to approximate the full GP. The focus of the comparisons will be on the quality of the selected kernel hyperparameters and the run time of the hyperparameter selection procedure.

The baselines are implemented with two popular GP packages: GPy [25] (implemented for CPU) and GPyTorch [20] (implemented for GPU). The spectral mixture (SM) kernel and spectral mixture product (SMP) kernel are used as the kernel functions. These give rise to eight different baselines: GPy-SM, GPy-SM-Sp, GPy-SMP, GPy-SMP-Sp, GPT-SM, GPT-SM-Sp, GPT-SMP, GPT-SMP-Sp, where we use "GPT" to denote "GPyTorch" and "Sp" to denote "SGPR". The default L-BFGS

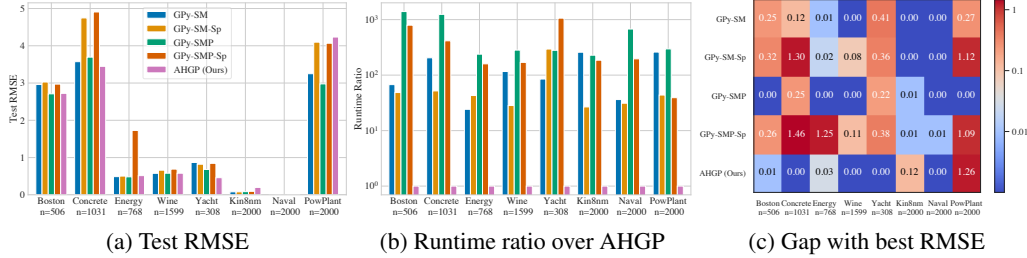

|                | (a) Test RMSE | (b) Runtime ratio over AHGP | (c) Gap with best RMSE |

Figure 2: Comparison of AHGP against the CPU baselines on regression benchmarks. In (c), the numbers are the differences of the corresponding method's test RMSE with the best RMSE on the respective dataset. Note that for Naval, the RMSEs are all very close to 0. (Only average test performance is shown here. Refer to Appendix C for complete results with error bars.)

optimizer is used for GPy and the default Adam [32] optimizer is used for GPyTorch. The GPyTorch baselines make use of batched conjugate gradient to invert the kernel matrix for efficient approximate inference. We additionally implement a full GP baseline with SMP kernel that uses Cholesky decomposition in PyTorch [48], and its MLL is optimized via reverse-mode automatic differentiation. We will refer to this baseline as PyT-AD-SMP.

**Experimental setup.** In our experiments, the training data are constructed by sampling multiple sets of synthetic input/output data from a GP prior with a stationary kernel. Dimensions vary from 2 to 15. More details about data generation are provided in Appendix C. A *single* neural model is trained on the synthetic data using Adam [32] with a fixed learning rate, and the same trained model is then used across all evaluations. To validate the effectiveness of our neural network model, we minimize the efforts of hyperparameter tuning during training. The only hyperparameters we tuned are learning rate and number of layers in $\mathrm{LocalTransformer}$ and $\mathrm{GlobalTransformer}$. Average pooling is used as the $\mathrm{AggregateFunction}$. Details about the hyperparameters used are included in Appendix C. After training is done with the synthetic data, the *same* trained neural model is directly used to predict the kernel hyperparameters for the various *unseen* GP use-cases that are shown later.

During evaluation and training, both the data input and output are standardized and the noise variance of GP is fixed at 0.01. Note that it is possible for our method to learn the noise variance too. In our experiments, we find the predictive performance is not sensitive to the noise variance if the data are standardized and setting it to 0.01 gives competitive performance for all baselines. Meanwhile, it is noted that spectral mixture kernel is flexible enough to model the noise variance component as well: if one of the Gaussian mixture components has weight $w$, $\mu = 0$ and very large $\sigma^2$, the corresponding kernel matrix will be approximately $wI$ and the weight $w$ represents the noise scale.

**Regression benchmarks.** We evaluate our method and the baselines on regression benchmarks from the UCI collection [1] used in Hernández-Lobato and Adams [29] and Sun et al. [66] following the same setup: the data are randomly split to 90% for training and 10% for testing. This splitting process is repeated 10 times and the average test performance is reported. Comparisons with CPU-based baselines on test RMSE, test log-likelihood and runtime are presented in Fig. 2 and Table 6 (Appendix C). We observe that AHGP has consistently lower run times than the baselines, averaging ~100 times faster. Nevertheless, the predictive performance of AHGP is comparable to (and sometimes better than) the strongest baselines, which perform MLL optimization without approximation. Notably, AHGP seems to perform slightly better on datasets with fewer data points, such as Yacht. We believe this demonstrates the robustness of AHGP when there is not enough data for MLL-opt based approaches to form reasonable point estimates. The sparse variational GP methods are faster than the full GP methods in general, but with lower performance on both test RMSE and test log-likelihood. Comparisons with GPU-based methods (GPyTorch-based and PyT-AD-SMP) are in Appendix C, where similar findings are obtained.

**Bayesian optimization.** Bayesian optimization [42] (BO) uses a GP as a surrogate model when the objective function is expensive to evaluate. The method involves fitting the GP kernel hyperparameters and maximizing an acquisition function to select a candidate point that is highly promising to achieve the function minima (maxima) under the model. Since the method is iterative, MLL optimization needs to be conducted at every BO iteration to update the GP. An amortized approach would greatly reduce the computation involved. We pick the best performing baselines on the regression benchmarks (GPy-SM, GPy-SM-Sp, PyT-AD-SMP) and compare them with AHGP. Expected improvement (EI)

is used as the acquisition function. We use standard test functions for global optimization [9] as the target functions for the BO experiments.

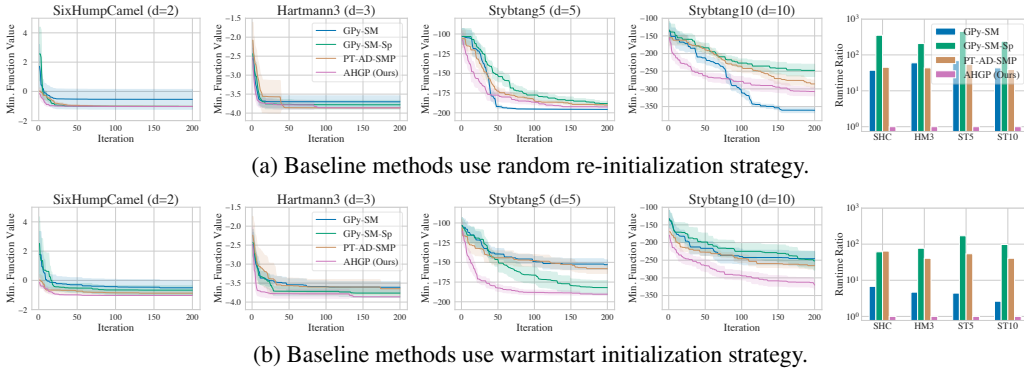

(a) Baseline methods use random re-initialization strategy.

(b) Baseline methods use warmstart initialization strategy.

Figure 3: BO performance comparisons. *Left*: Minimum function values found v.s. number of BO iterations. Shaded region represents 0.5 standard deviation over 10 runs. *Right*: Runtime ratio over AHGP. (Only average is plotted here, refer to Appendix C for mean and standard deviation.)

At the start of every BO iteration, the hyperparameters are randomly re-initialized for all baselines. A sample of the experimental results is shown in Fig. 3a. (Full results can be found in Appendix C.) Again, AHGP is a substantial improvement in run-time. In terms of minimum values found, AHGP is on par with the baselines on some functions and slightly worse on functions with higher dimensionality. Of particular note—consistent with what was seen on the regression benchmarks—AHGP has the greatest improvement in the beginning when there are few observations available.

To ensure fairness to baseline fitting procedures, we also conducted experiments where the hyperparameter selection in the BO inner loop was initialized using the best from the previous iteration. As expected, this warm-starting results in decreased run times for the baselines, although still slower than AHGP (in Fig. 3b). This warm-starting, however, seems to compromise the hyperparameter selection—presumably due to local minima—and damage the overall outer loop optimization.

**Bayesian quadrature.** Bayesian quadrature [46, 52] performs Bayesian inference about the value of a difficult numerical integral through modeling the underlying function as a GP. The tractability of the Gaussian process makes it relatively easy to calculate a Bayesian estimate of many integrals in closed form. For example, Gunter et al. [27] applies Bayesian quadrature to the task of probabilistic inference and achieves faster convergence than Monte Carlo methods.

| TARGET FUNCTION | GPy-SM | GPy-SM-Sp | PyT-AD-SMP | **AHGP(Ours)** |
|---|---|---|---|---|
| Hennig1D | NaN | -29.66±26.12 | **4.23±0.16** | 3.89±0.27 |
| Hennig2D | NaN | -39.99±36.57 | -0.13±4.48 | **2.41±2.30** |
| Sombrero2D | NaN | -0.63±3.56 | **4.95±0.26** | 3.23±0.43 |
| Circular Gaussian | 3.79±1.01 | 4.01±1.27 | **5.17±0.04** | 3.47±1.09 |

Table 1: Log-likelihood of the true integral evaluated at the final prediction of Bayesian quadrature.

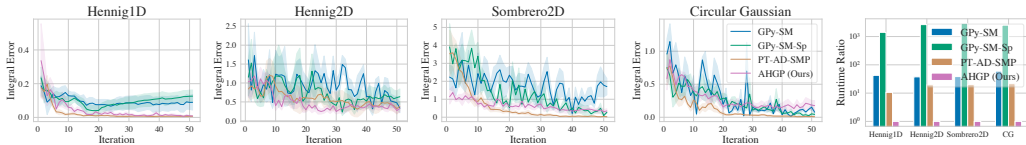

Figure 4: Bayesian quadrature performance comparisons. *Left*: Integral error v.s. number of BQ iterations. Shaded region represents 0.5 standard deviation over 5 runs. *Right*: Runtime ratio over AHGP.

We evaluate our model on four standard test functions provided in Emukit [47]. In Fig. 4, we plot the average distance to ground truth versus number of iterations. Table 1 shows the log-likelihood of the true integral evaluated at the final probabilistic estimate. AHGP compares well with the baselines across all tasks. More importantly, AHGP appears to be robust across different tasks while the baselines often suffer from large variances and occasional divergence.

# 6    Related Work

**Amortized variational inference.**    Amortized variational inference [33, 34, 26, 56] was a major inspiration for the present work. Kingma and Welling [33] first proposed to use a neural network to amortize the expensive variational inference optimization through training on the evidence lower bound (ELBO) with the reparametrization trick. Our work is motivated by this idea, where we use a neural network to produce an estimate of the kernel hyperparameters conditioned on the data. Instead of producing approximate posteriors, we produce point estimates. Note that our model can be directly extended to the variational inference framework, which we leave as future work.

**Neural processes.**    Another line of work, *neural processes (NPs)* [22, 21, 31, 40], makes use of amortized variational inference to predict a distribution over regression functions from a set of observed input-output pairs. Our work targets estimation of GP hyperparameters as part of a full GP inference procedure, while neural processes serve as an end-to-end approximations to the GP inference itself. It should also be noted that our method is task-agnostic: one single trained neural model can be used across different problems with different dimensionality. In the neural process approach, different neural networks need to be trained for different types of tasks. In particular, Kim et al. [31] proposed to extend the original NP by introducing attention mechanism between the context data and the target data for better modeling the underlying structure of the problem. In comparison, the hierarchical attention network proposed in our method models not only between data points but also between dimensions, which allows the task-agnostic nature of our method.

**GPU-accelerated GP inference.**    There have been recent efforts [20, 71] to scale up GP inference with the help of GPUs by introducing smartly-designed batched conjugate gradient algorithms. In contrast, our work focuses on providing a task-agnostic method for amortizing the cost of GP model selection. Note that AHGP is also able to be deployed on GPUs.

**Meta-learning.**    Under the general meta-learning framework, our model could be regarded as a meta-model for predicting hyperparameters. There have been recent explorations of meta-learning that are model-based [58, 11], metric based [69, 35] or optimization based [59, 2, 18]. Gordon et al. [24] extends the meta-learning framework to probabilistic inference for regression. To better warm-start BO through leveraging previous related BO runs, meta-learning has also been proposed for BO recently [14, 49, 15]. The BO meta-learning approaches focus on the few-shot learning setup: i.e. how to build a good surrogate model with very few function evaluations for a new but similar task. In comparison, our method focuses on fast inference of GP hyperparameters, and our single trained model is adaptation-free and can be directly applied to a wider range of new GP use cases.

# 7    Conclusions and Future Work

We introduced *amortized hyperparameter inference for Gaussian processes* (AHGP). The proposed neural model is not only versatile to accommodate tasks of different size and dimensionality but also permutation invariant of both data and dimensions. We experimentally show that a single amortized inference model trained on synthetic data is able to directly generalize to different unseen real-world GP use cases. This method is capable of producing hyperparameters that are comparable in quality to those from the conventional MLL maximization approaches, while being on average $\sim 100$ times faster. One limitation of this approach, however, is that training on very large datasets consumes too much GPU memory or becomes computationally expensive due to the kernel matrix size and the cost of inverting the matrix. One direction of future work is to develop methods that are more efficient both in memory and computation.

# Broader Impact

Iterative hyperparameter optimization procedures often place a heavy computation burden on people who apply Gaussian processes to real-world applications. The optimization procedure itself usually has hyperparameters to be tuned (learning rate, number of iterations, etc.), which further increases the computational cost. Our proposed method amortizes this cost by training a single meta-model that is then useful across a wide range of tasks. Once the meta-model is trained, it can be repeatedly applied to future kernel hyperparameter selection tasks, reducing resource usage and carbon footprint.

The minimal computation required by our method also makes it more accessible to the general public instead of only to those with abundant computing resources.

Like most deep learning models, our neural model has the potential risk of overfitting and low robustness. In an effort to avoid this, we use only synthetic data generated from a family of kernel function space that is expressive enough to cover a variety of Gaussian process use cases. Our goal is to avoid biasing the model towards any particular task. Additionally, we impose regularizations such as permutation invariance and weight sharing to encourage generalizable representations. Even with all these efforts, our model might still produce misspecified hyperparameters which can lead to poor prediction performance versus conventional MLL optimization procedures.

## Acknowledgments and Disclosure of Funding

We would like to thank Yucen Luo and members of the Princeton Laboratory for Intelligent Probabilistic Systems for valuable discussions and feedback. Funding: This work was partially funded by NSF IIS-2007278 and a Princeton SEAS Innovation Grant. Other financial interests: RPA is on the board of directors at Cambridge Machines Ltd. and a scientific advisor to Manifold Bio.

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
