[Supplementary Material]

# Appendix

## A   Attention Mechanism

In this section, we give a brief overview of the most commonly used attention mechanisms.

*Dot-product attention* is an attention mechanism that takes in $m$ queries $\mathbf{Q} \in \mathbb{R}^{m \times d_k}$ and maps them to outputs using $n$ key-value pairs $\mathbf{K} \in \mathbb{R}^{n \times d_k}$, $\mathbf{V} \in \mathbb{R}^{n \times d_k}$. The output is a weighted sum of values with each value's weight being determined by the dot product of the corresponding key with the query:

$$\text{Attention}(\mathbf{Q}, \mathbf{K}, \mathbf{V}) = \mathbf{WV} \in \mathbb{R}^{m \times d_v}, \text{with } \mathbf{W} = \text{softmax}(\mathbf{Q}\mathbf{K}^T / \sqrt{d_k}) \in \mathbb{R}^{m \times n}.$$

Intuitively, for each query, attention computes an aggregated vector based on its relevance to the key-value pairs.

*Multi-head attention*, first introduced in [68], extends a single dot-product attention to multiple attentions. It first projects the original keys, values and queries onto $H$ different keys, values and queries. And dot attention is applied for each of these $H$ projections. The final values are calculated by taking a linear transformation of the concatenated different head-specific values:

$$\text{MultiHead}(\mathbf{Q}, \mathbf{K}, \mathbf{V}) := \text{concat}\left(\text{head}_1, \ldots, \text{head}_H\right) \mathbf{W},$$

$$\text{head}_h := \text{Attention}(\mathbf{Q}\mathbf{W}_h^{\mathbf{Q}}, \mathbf{K}\mathbf{W}_h^{\mathbf{K}}, \mathbf{V}\mathbf{W}_h^{\mathbf{V}}).$$

The multiple heads allow the attention mechanism to learn richer representations by combining representations from different subspaces.

For sequential data, *self-attention* defines a key-value-query tuple for each datum in a sequence so that they can attend to each other. Combined with multi-head attention, one could apply a feed-forward network right after with residual connection to make up a multi-head self-attention sub-block. By stacking up multiple multi-head self-attention sub-blocks, one could formulate a multi-layer *Transformer* encoder [68] that allows extracting feature representations hierarchically through modeling interactions between input representations at different levels and maps a sequence of data to a sequence of final feature representations for downstream tasks. It is most commonly used as the fundamental functional block of the powerful Transformer type of models that achieves state-of-the-art performance on many NLP tasks [68, 10].

## B   Proof of Proposition 1

We start by defining permutation equivariant and permutation invariant functions.

**Definition 1.** *Let $S_n$ be the set of all permutations of indices $\{1, 2, \cdots, n\}$. A function $f : \mathcal{X}^n \to \mathcal{Y}^n$ is **permutation equivariant** if and only if for any permutation $\pi \in S_n$, $f(\pi x) = \pi f(x)$.*

**Definition 2.** *Let $S_n$ be the set of all permutations of indices $\{1, 2, \cdots, n\}$. A function $f : \mathcal{X}^n \to \mathcal{Y}$ is **permutation invariant** if and only if for any permutation $\pi \in S_n$, $f(\pi x) = f(x)$.*

Next, we start the proof.

*Proof.* We assume our neural network takes in the $l$-th dataset $\mathcal{D}^{(l)}$ as input. By definition of the multi-head self-attention mechanism, it is obvious that a single multi-head self-attention sublock is permutation equivariant. From [76] we know stacks of permutation equivariant layers are still permutation equivariant. Therefore, for every dimension, the input/output data $\{(\mathbf{x}_{i,d}^{(l)}, y_i^{(l)})\}_{i=1}^{N_l}$ are always mapped to the same corresponding $\{\mathbf{h}_{i,d}^{(l)}\}_{i=1}^{N_l}$ regardless of their order. Further if AggregateFunction is permutation invariant, we have $\mathbf{h}_{\text{local},d}^{(l)}$ invariant of the order of data points for each dimension.

As the LocalTransformer is sharing the same weights for all dimensions, $\{\mathbf{h}_{\text{local},d}^{(l)}\}_{d=1}^{D_l}$ will remain equivariant with regards to permutation of dimensions. Next $\{\mathbf{h}_{\text{local},d}^{(l)}\}_{d=1}^{D_l}$ are passed through stacks of multi-head self-attention subblocks which is permutation equivariant, therefore the final $\{\mathbf{h}_{\text{global},d}^{(l)}\}_{d=1}^{D_l}$ are permutation equivariant. Again the weight sharing of the final MLP ensures

that the final predicted spectral density hyperparameters $\{\theta_d^{(l)}\}_{d=1}^{D_l}$ are permutation equivariant with regards to data dimensions. They are also permutation invariant with regards to data points, as $\{\mathbf{h}_{\text{local},d}^{(l)}\}_{d=1}^{D_l}$ are invariant of the order of input/output data. $\qquad\square$

## C   Experimental Details and Results

### C.1   Synthetic Training Dataset Generation

In our experiments, our training dataset is constructed by sampling multiple sets of input/output data from synthetically generated GPs with stationary kernel functions. For each GP, its data dimensionality is sampled uniform randomly from 2~15. To sample flexible kernel functions, we randomly generate mixtures of Gaussians to represent its spectral density. The weights of the Gaussian mixtures are drawn from Dirichlet distribution and the lengthscales (i.e., $1/\sqrt{2}\pi\sigma_{(d),m}$'s) are sampled from a log-uniform distribution. The number of mixtures is set to 10 and the concentration parameter of the Dirichlet distribution are set to 1. The input of the data points $\{\mathbf{x}_i^{(l)}\}_{i=1}^{N_l}$ are generated from a Poisson point process within the hypercube $[-1,1]^{D_l}$ with average density being 30. The data output values $\{y_i^{(l)}\}_{i=1}^{N_l}$ are generated from priors of the GP with its specified kernel function. The observation noise is i.i.d. randomly sampled from $\mathcal{N}(0, 0.01)$. We generate 10000 sets of input/output data to be used as the whole dataset, of which we do a split with 50% used for training and the rest for validation. It is worth noting that although our AHGP neural model is trained on datasets with relative small size ($\sim$30), the trained neural model is able to generalize on real-world datasets of much larger size, which is presented in details in Section 5. This also shows the effectiveness of the inductive bias introduced in our hierarchical attention network.

### C.2   Training Details

Our model is trained with PyTorch using the Adam [32] optimizer with a fixed learning rate and a batch size of 64. A description of our model architecture is provided in Table 2, 3. We also apply 0.1 dropout to self-attention and the MLPs. The number of Gaussian mixtures in the spectral density prediction is fixed at 10. To validate the effectiveness of our neural network model, we minimize the efforts of hyperparameter tuning during training. The only hyperparameters we tuned are learning rate ($\{10^{-3}, 10^{-4}, 10^{-5}, 10^{-6}\}$) and number of layers in LocalTransformer and GlobalTransformer (2~8). The hyperparameters are tuned based on performance on the validation set.

| Embed each $(\mathbf{x}_{i,d}^{(l)}, y_i^{(l)})$ to 256 dim |
| :---: |
| Transformer sublayer (4 heads, representation dim 256, feedforward dim 512) |
| Transformer sublayer (4 heads, representation dim 256, feedforward dim 512) |
| Transformer sublayer (4 heads, representation dim 256, feedforward dim 512) |
| Transformer sublayer (4 heads, representation dim 256, feedforward dim 512) |
| Transformer sublayer (4 heads, representation dim 256, feedforward dim 512) |
| Transformer sublayer (4 heads, representation dim 256, feedforward dim 512) |
| Transformer sublayer (4 heads, representation dim 256, feedforward dim 512) |
| Transformer sublayer (4 heads, representation dim 256, feedforward dim 512) |
| AggregateFunction: Average pooling |

Table 2: LocalTransformer architecture and AggregateFunction

Since spectral mixture kernel assume the variance of the kernel function is normalized, we standardize the function values $\{y_i^{(l)}\}_{i=1}^{N_l}$. We also standardize the data input $\{\mathbf{x}_i^{(l)}\}_{i=1}^{N_l}$ as a standard procedure. During training, we find the validation performance is most sensitive to learning rate and increasing layers of the Transformers slightly helps. The final model is trained for 200 epochs with batch size 64, learning rate $10^{-5}$ and 8 layers in both Local and Global Transformer. The training is done on an NVIDIA GTX 1080 Ti GPU. All the evaluation experiments are run using one core of an Intel(R) Core(TM) i7-6850K CPU for CPU runtime comparisons and an NVIDIA GTX 1080 Ti GPU for GPU runtime comparisons.

| | |
|---|---|
| Transformer sublayer (4 heads, representation dim 256, feedforward dim 512) | |
| Transformer sublayer (4 heads, representation dim 256, feedforward dim 512) | |
| Transformer sublayer (4 heads, representation dim 256, feedforward dim 512) | |
| Transformer sublayer (4 heads, representation dim 256, feedforward dim 512) | |
| Transformer sublayer (4 heads, representation dim 256, feedforward dim 512) | |
| Transformer sublayer (4 heads, representation dim 256, feedforward dim 512) | |
| Transformer sublayer (4 heads, representation dim 256, feedforward dim 512) | |
| Transformer sublayer (4 heads, representation dim 256, feedforward dim 512) | |
| MLP for predicting weights, means and variances (each with hidden dim [256, 128]) | |

Table 3: GlobalTransformer architecture and the final MLP

## C.3 Is the Model Actually Learning?

Our method (AHGP) has demonstrated competitive performance across different applications. How-ever, is the neural model really learning? In this section, we will construct experiments to conduct a sanity-check on this empirically. In the experiments, we augment each dataset by adding a new feature dimension to each datapoint. We test with two extreme cases: 1. an i.i.d random Gaussian noise and 2. the exact same label of the data point. The new dimension is also standardized and treated just like normal features. And then we apply our trained neural model on the augmented regression datasets.

One should expect a sensible model to predict the noise dimension with large lengthscales, as the dimension does not contain any useful information and should be discarded by assigning large lengthscales. Meanwhile, the label dimension should be identified with the smallest lengthscales because they carry the most information across all dimensions. Table 4 shows a summary of the weighted lengthscales predicted by our model on the regression benchmarks. Results show that our model is able to consistently identify the noise dimension with very large lengthscales and the label dimension with the smallest lengthscales, which suggests that our model is learning generalizable representations.

| DATASET | Boston | Concrete | Energy | Wine | Yacht | Kin8nm | Naval | PowPlant |
|---|---|---|---|---|---|---|---|---|
| Noise dim. len. | 163.14±1.61 | 185.74±3.88 | 180.71±0.91 | 87.31±3.84 | 217.94±4.67 | 129.77±1.81 | 148.88±0.59 | 118.65±2.46 |
| Label dim. len. | 1.77±0.00 | 1.71±0.00 | 1.51±0.00 | 1.40±0.01 | 1.15±0.01 | 1.46±0.00 | 1.37±0.00 | 1.63±0.00 |

Table 4: Predicted weighted lengthscales of noise and label dimensions

## C.4 Regression Benchmarks

For SGPR methods, the number of inducing points is set to $10\%$ of the number of training data. The full comparisons with the CPU-based baselines in terms of test RMSE, test log marginal likelihood and hyperparameter selection runtime are presented in Table 5, 6, 7.

| DATASET | GPy-SM | GPy-SM-Sp | GPy-SMP | GPy-SMP-Sp | AHGP(Ours) |
|---|---|---|---|---|---|
| Boston | 2.960±0.506 | 3.027±0.506 | **2.710±0.482** | 2.971±0.487 | 2.723±0.387 |
| Concrete | 3.573±0.848 | 4.747±0.386 | 3.695±0.805 | 4.907±0.393 | **3.448±0.448** |
| Energy | 0.488±0.036 | 0.503±0.058 | **0.483±0.044** | 1.728±1.233 | 0.515±0.071 |
| Wine | **0.577±0.027** | 0.661±0.028 | 0.580±0.028 | 0.690±0.038 | 0.580±0.035 |
| Yacht | 0.868±0.567 | 0.821±0.283 | 0.681±0.288 | 0.845±0.387 | **0.460±0.265** |
| Kin8nm | **0.080±0.003** | 0.080±0.004 | 0.080±0.004 | 0.085±0.004 | 0.199±0.008 |
| Naval | **0.000±0.000** | 0.001±0.001 | 0.000±0.000 | 0.006±0.002 | 0.002±0.000 |
| PowPlant | 3.250±0.299 | 4.097±0.348 | **2.977±0.210** | 4.068±0.221 | 4.234±0.242 |

Table 5: Test RMSE on regression benchmarks: CPU-based methods

The comparisons of AHGP with the GPU-based baselines are presented in Fig. 5 and Table 8, 9, 10. Results are similar to comparisons with the CPU-based baselines in the main section. Note that GPy-

| DATASET | GPy-SM | GPy-SM-Sp | GPy-SMP | GPy-SMP-Sp | AHGP (Ours) |
|---|---|---|---|---|---|
| Boston | -2.649±0.364 | -3.527±1.086 | -2.538±0.281 | -2.939±0.642 | **-2.367±0.115** |
| Concrete | **-2.656±0.637** | -3.435±0.238 | -2.690±0.574 | -3.757±0.579 | -3.460±1.334 |
| Energy | -1.059±0.019 | -1.103±0.046 | -1.079±0.028 | -2.534±0.557 | **-0.837±0.215** |
| Wine | -0.427±0.058 | -4.331±1.328 | -0.410±0.053 | -1.053±0.051 | **-0.321±0.075** |
| Yacht | -1.573±0.254 | -1.526±0.108 | -1.500±0.077 | -1.979±0.742 | **-0.997±0.092** |
| Kin8nm | **1.119±0.044** | 0.655±0.177 | 1.020±0.054 | 0.748±0.198 | 0.192±0.039 |
| Naval | 6.157±0.267 | 5.433±1.063 | **6.211±0.011** | 3.643±0.164 | 5.393±0.102 |
| PowPlant | -2.591±0.115 | -3.831±0.330 | **-2.475±0.060** | -2.936±0.047 | -3.112±0.151 |

Table 6: Test log-likelihood on regression benchmarks: CPU-based methods

| DATASET | GPy-SM | GPy-SM-Sp | GPy-SMP | GPy-SMP-Sp | AHGP(Ours) |
|---|---|---|---|---|---|
| Boston | 83.56±65.67 | 60.16±9.98 | 1731.96±525.13 | 984.17±282.21 | **1.24±0.16** |
| Concrete | 545.08±195.05 | 138.01±36.99 | 3307.03±1484.01 | 1103.31±68.64 | **2.66±0.05** |
| Energy | 43.05±16.51 | 76.30±5.50 | 424.39±124.63 | 284.02±376.79 | **1.78±0.30** |
| Wine | 1629.73±796.54 | 397.35±163.41 | 3961.91±1617.62 | 2380.69±885.05 | **14.04±1.59** |
| Yacht | 18.68±8.44 | 64.87±1.12 | 61.56±26.95 | 233.75±157.30 | **0.22±0.01** |
| Kin8nm | 3391.64±958.03 | 350.49±69.52 | 2999.11±1036.25 | 2435.50±169.28 | **13.13±0.84** |
| Naval | 915.49±881.60 | 786.15±163.73 | 17070.97 ± 5960.81 | 4955.79 ± 2251.73 | **25.25±0.57** |
| PowPlant | 1869.34±1616.23 | 314.33±74.13 | 2131.19±490.66 | 281.75±207.83 | **7.19±0.55** |

Table 7: Runtime (in seconds) on regression benchmark: CPU-based methods

Torch baselines perform slightly worse than GPy baselines, since GPyTorch uses conjugate gradient method to approximately solve for matrix inverse instead of doing the Cholesky decomposition. In comparison, PyT-AD-SMP uses Cholesky decomposition and generally achieves better performance than GPT-SMP.

(a) Test RMSE  (b) GPU runtime ratio over AHGP  (c) Gap with best RMSE

Figure 5: Comparison of AHGP against the GPU-based baselines on regression benchmarks. In (c), the numbers are the differences of the corresponding method's test RMSE with the best RMSE on the respective dataset. Note that for Naval, the RMSEs are all very close to 0 except PyT-AD-SMP which runs out of GPU memory.

| DATASET | GPT-SM | GPT-SM-Sp | PyT-AD-SMP | GPT-SMP | GPT-SMP-Sp | AHGP(Ours) |
|---|---|---|---|---|---|---|
| Boston | 3.150±0.674 | 4.582±1.168 | 2.906±0.488 | 3.294±0.900 | 4.647±1.133 | **2.723±0.387** |
| Concrete | 6.295±1.071 | 7.568±0.558 | 3.710±0.742 | 6.393±0.877 | 7.290±0.535 | **3.448±0.448** |
| Energy | 0.485±0.041 | 1.004±0.166 | **0.472±0.041** | 0.483±0.050 | 0.958±0.201 | 0.515±0.071 |
| Wine | 0.641±0.030 | 0.663±0.033 | **0.579±0.027** | 0.644±0.030 | 0.666±0.034 | 0.580±0.035 |
| Yacht | 0.678±0.315 | 2.597±1.149 | 0.622±0.265 | 0.681±0.316 | 2.514±1.539 | **0.460±0.265** |
| Kin8nm | 0.089±0.005 | 0.108±0.004 | **0.081±0.004** | 0.089±0.005 | 0.105±0.003 | 0.199±0.008 |
| Naval | **0.000±0.000** | 0.003±0.000 | Out of memory | 0.001±0.000 | 0.003±0.000 | 0.002±0.000 |
| PowPlant | 4.501±0.238 | 4.412±0.241 | **2.922±0.256** | 4.462±0.230 | 4.161±0.241 | 4.234±0.242 |

Table 8: Test RMSE on regression benchmarks: GPU-based methods

| DATASET | GPT-SM | GPT-SM-Sp | PyT-AD-SMP | GPT-SMP | GPT-SMP-Sp | AHGP (Ours) |
|---------|--------|-----------|------------|---------|------------|-------------|
| Boston | -2.692±0.442 | -12.440±5.930 | -2.757±0.416 | -2.687±0.439 | -11.390±5.270 | **-2.367±0.115** |
| Concrete | **-3.030±0.640** | -3.347±0.107 | -3.132±1.262 | -3.095±0.720 | -3.285±0.078 | -3.460±1.334 |
| Energy | -1.073±0.020 | -1.269±0.064 | -1.496±0.400 | -1.068±0.023 | -1.266±0.067 | **-0.837±0.215** |
| Wine | -0.517±0.079 | -1.067±0.129 | **-0.306±0.064** | -0.527±0.076 | -1.028±0.115 | -0.321±0.075 |
| Yacht | -1.492±0.105 | -2.416±0.634 | -1.030±0.770 | -1.492±0.106 | -2.477±1.071 | **-0.997±0.092** |
| Kin8nm | 0.921±0.086 | 0.819±0.042 | **1.108±0.047** | 0.917±0.090 | 0.854±0.045 | 0.192±0.039 |
| Naval | 5.892±0.028 | 5.253±0.215 | Out of memory | **5.933±0.017** | 5.372±0.113 | 5.393±0.102 |
| PowPlant | -2.923±0.120 | -2.903±0.135 | **-2.540±0.153** | -2.920±0.150 | -2.834±0.078 | -3.112±0.151 |

Table 9: Test log-likelihood on regression benchmarks: GPU-based methods

| DATASET | GPT-SM | GPT-SM-Sp | PyT-AD-SMP | GPT-SMP | GPT-SMP-Sp | AHGP(Ours) |
|---------|--------|-----------|------------|---------|------------|------------|
| Boston | 10.99±0.05 | 4.80±0.04 | 4.19±1.16 | 10.33±0.07 | 8.83±0.47 | **0.05±0.00** |
| Concrete | 32.10±3.69 | 3.98±0.14 | 4.12±0.02 | 32.69±10.72 | 8.24±0.14 | **0.13±0.00** |
| Energy | 4.17±0.12 | 2.47±0.13 | 6.49±1.76 | 6.72±0.13 | 6.17±0.56 | **0.08±0.02** |
| Wine | 46.63±0.19 | 5.32±0.15 | 10.20±0.08 | 39.19±0.07 | 11.22±0.31 | **0.38±0.00** |
| Yacht | 2.37±0.16 | 4.01±0.16 | 1.64±0.05 | 2.53±0.15 | 9.96±0.20 | **0.02±0.00** |
| Kin8nm | 66.55±21.34 | 10.86±0.14 | 10.80±0.15 | 53.79±1.13 | 19.23±0.69 | **0.46±0.00** |
| Naval | 99.58±1.98 | 32.12±0.70 | Out of memory | 106.01±0.52 | 46.83±0.57 | **0.84±0.13** |
| PowPlant | 49.36±0.14 | 11.29±0.67 | 7.66±0.06 | 32.17±0.06 | 14.15±0.89 | **0.33±0.01** |

Table 10: Runtime (in seconds) on regression benchmark: GPU-based methods

#### C.4.1 Increasing the number of inducing points on sparse variational GP

To fully compare the sparse variational GP (SGPR) with our method, we conducted another experiment by varying the number of inducing points used in SGPR. To illustrate the difference when different number of inducing points is used, we pick the dataset *Concrete* on which we see an obvious performance gap between GPy-SM (MLL-opt) and GPy-SM-Sp (SGPR). In the experiment, the number of inducing points in GPy-SM-Sp is set to $\{10\%, 20\%, 40\%, 60\%, 80\%, 100\%\}$ of the total number of training data. The comparisons with GPy-SM and AHGP in terms of test RMSE, test log-likelihood and run time is shown in Fig. 6 and Table 11. By increasing the number of inducing points, the gap between SGPR and MLL-opt is gradually reduced, although a small gap remains even when the number of inducing points is set to $100\%$. We also observe increased run times when then number of inducing points is increased. As SGPR has a more complicated optimization problem than MLL-opt, its run time is slower than MLL-opt when the number of inducing points is set to the same as the training data.

(a) Test RMSE  (b) CPU runtime ratio over AHGP

Figure 6: Comparison of SGPR (GPy-SM-Sp) with MLL-opt (GPy-SM) and AHGP. The percentage represents how many inducing points (in terms of the number of training data) are used in SGPR.

#### C.4.2 Why does AHGP sometimes have better predictive performance?

Surprisingly, AHGP performs even better than the MLL-opt approaches on dataset with fewer data points (e.g., Yacht with $n = 308$) in terms both test RMSE and test log-likelihood. It is therefore of great interest to investigate the underlying reason behind this phenomenon. One natural question to ask would be: "Is AHGP learning a better kernel function than the optimized one on the training

| METHOD | Test RMSE ↓ | Test log-likelihood ↑ | Runtime ↓ |
|---|---|---|---|
| GPy-SM-Sp-10% | 4.747±0.386 | -3.435±0.238 | 138.01±36.99 |
| GPy-SM-Sp-20% | 4.277±0.268 | -3.155±0.135 | 261.36±68.15 |
| GPy-SM-Sp-40% | 3.988±0.332 | -2.999±0.184 | 620.73±118.55 |
| GPy-SM-Sp-60% | 3.878±0.464 | -2.870±0.297 | 876.00±157.93 |
| GPy-SM-Sp-80% | 3.818±0.614 | -2.699±0.175 | 1144.39±111.37 |
| GPy-SM-Sp-100% | 3.704±0.396 | -2.690±0.144 | 1818.10±190.88 |
| GPy-SM | 3.573±0.848 | **-2.656±0.637** | 545.08±195.05 |
| **AHGP(Ours)** | **3.448±0.448** | -3.460±1.334 | **2.66±0.05** |

Table 11: Comparison of test RMSE, test log-likelihood and runtime

data? Or is it some other reason?". We examine the marginal likelihood on the training data achieved by AHGP and MLL-opt. It turns out the MLL-opt method still achieves a slightly better marginal likelihood on the training data (1.04 v.s. 0.85). This demonstrates that MLL-opt method is doing a good job in optimizing the marginal likelihood. However, its performance on test set shows that a perfectly optimized kernel hyperparameter on a relative small number of training data can still lead to overfitting. In comparison, AHGP only uses synthetic data during training and this training procedure seems to add extra regularization effect which prevents kernel hyperparamter estimate of AHGP from overfitting towards the training data and hence helps AHGP achieve a better performance on the in test time.

## C.5 Bayesian Optimization

We pick the best performing baselines on the regression benchmarks (GPy-SM, GPy-SM-Sp, PyT-AD-SMP) and compare them with our method. The number of inducing points in GPy-SM-Sp is set to 20. Standard test functions for global optimization [9] are used as the target functions for Bayesian optimization. Five initial input points are randomly sampled and function values at the points are evaluated; those input points with their function values serve as the initial input/output data of GP. At the beginning of each BO iteration, hyperparameters are randomly reinitialized for all the baselines and then optimized through MLL optimization. The full results are shown in Fig. 7, 8a and Table 12.

For comparison, we also implement another warmstart initialization strategy which sets the initialization as the best hyperparameters from the previous BO iteration. The full results are shown in Fig. 9, 8b and Table 13.

Figure 7: Bayesian optimization performance comparison: random initialization strategy. Shaded region indicates 0.5 standard deviations over 10 runs.

(a) Random initialization strategy.

(b) Warmstart initialization strategy.

Figure 8: Runtime on Bayesian optimization tasks.

| FUNCTION | GPy-SM | GPy-SM-Sp | PyT-AD-SMP | **AHGP(Ours)** |
|---|---|---|---|---|
| Ackley | 2275.32±343.28 | 5871.42±489.04 | 1288.16±70.09 | **47.86±3.66** |
| Griewank | 2192.96±1003.29 | 5863.13±626.95 | 1272.42±76.59 | **46.36±2.84** |
| Hartmann3 | 614.48±169.35 | 5801.73±559.05 | 711.35±7.47 | **16.44±1.44** |
| Hartmann6 | 1721.76±568.42 | 5935.85±503.19 | 961.31±24.82 | **28.65±2.44** |
| Levy | 3903.18±483.52 | 5707.24±628.55 | 1282.01±70.42 | **45.04±5.59** |
| SixHumpCamel | 1127.34±466.70 | 5898.18±608.69 | 625.29±4.76 | **12.84±1.66** |
| Michalewicz10 | 1115.01±152.13 | 5891.81±461.76 | 1295.83±73.77 | **48.48±4.24** |
| Michalewicz2 | 383.41±62.54 | 5762.95±486.45 | 629.56±4.56 | **13.10±1.57** |
| Stybtang10 | 2826.95±581.05 | 5849.69±526.99 | 1276.52±77.09 | **50.12±3.14** |
| Stybtang2 | 886.45±424.75 | 5776.96±588.76 | 633.21±10.43 | **12.80±1.63** |
| Stybtang5 | 1119.24±186.60 | 6186.69±562.42 | 872.66±19.39 | **25.74±2.87** |

Table 12: Runtime (in seconds) on Bayesian optimization tasks: random initialization strategy

Figure 9: Bayesian optimization performance comparison: warmstart initialization strategy. Shaded region indicates 0.5 standard deviations over 10 runs.

To evaluate our method on real-world Bayesian optimization problems, we additionally applied our method to tuning learning and model hyperparameters of logistic regression via BO. The goal is to find the hyperparameters that achieves the highest test accuracy when a fixed amount of time is used for training. We experiment on the task of training logistic regression on MNIST data with stochastic gradient descent. The training involves four hyperparameters: learning rate, $\ell_2$ regularization parameter, $\ell_1$ regularization parameter and mini-batch size. We present the results in Fig. 10 and Table 14. AHGP achieves comparable test accuracy with the strongest baselines

| FUNCTION | GPy-SM | GPy-SM-Sp | PyT-AD-SMP | **AHGP(Ours)** |
|---|---|---|---|---|
| Ackley | 92.12±49.74 | 4381.99±2374.99 | 1933.38±517.41 | **51.35±2.11** |
| Griewank | 115.26±45.74 | 4066.03±1975.20 | 1998.14±864.66 | **49.59±5.81** |
| Hartmann3 | 91.61±37.23 | 1501.48±1299.30 | 792.04±74.54 | **19.51±0.98** |
| Hartmann6 | 82.10±35.08 | 3957.00±2495.70 | 1853.98±751.37 | **31.47±0.92** |
| Levy | 114.29±60.44 | 5024.81±1751.92 | 2022.96±461.09 | **49.18±5.91** |
| SixHumpCamel | 101.05±66.46 | 912.45±634.74 | 957.04±375.30 | **14.90±0.44** |
| Michalewicz10 | 103.03±64.24 | 4005.42±2546.93 | 2133.61±1378.63 | **51.78±1.84** |
| Michalewicz2 | 47.00±19.18 | 311.37±209.75 | 622.09±36.82 | **15.16±0.42** |
| Stybtang10 | 126.53±109.21 | 4692.10±2186.68 | 1952.56±806.56 | **48.05±5.44** |
| Stybtang2 | 119.43±105.15 | 969.05±945.01 | 734.70±219.21 | **14.38±0.26** |
| Stybtang5 | 120.84±64.67 | 4655.30±1802.18 | 1482.06±504.21 | **27.35±1.28** |

Table 13: Runtime (in seconds) on Bayesian optimization tasks: warmstart initialization strategy

(GPy-SM, PyT-AD-SMP) while being much faster. The SGPR baseline, however, has a lower test accuracy and much longer runtime.

Figure 10: Bayesian optimization for training logistic regression on MNIST.

| GPy-SM | GPy-SM-Sp | PyT-AD-SMP | **AHGP(Ours)** |
|---|---|---|---|
| 195.90 ± 99.81 | 1067.70±41.40 | 23.85±0.45 | **1.23±0.06** |

Table 14: Runtime (in seconds) on Bayesian optimization for training logistic regression on MNIST

## C.6 Bayesian Quadrature

For all the tasks, we start with 10 initial points and their function values. The function values are standardized. And the results are reported on the standardized function values. Random initialization strategy is used for GP hyperparameters at the start of each outer iteration. The number of inducing points in GPy-SM-Sp is set to 20. The full runtime results are reported in Table 15.

| FUNCTION | GPy-SM | GPy-SM-Sp | PyT-AD-SMP | **AHGP(Ours)** |
|---|---|---|---|---|
| Circular Gaussian | 1.59±0.68 | 66.90±23.00 | 0.57±0.03 | **0.03±0.00** |
| Hennig1D | 0.90±0.17 | 29.77±19.83 | 0.22±0.01 | **0.02±0.00** |
| Hennig2D | 1.03±0.19 | 72.47±26.82 | 0.54±0.05 | **0.03±0.00** |
| Sombrero2D | 1.02±0.26 | 75.42±20.06 | 0.53±0.02 | **0.03±0.00** |

Table 15: Runtime (in seconds) on Bayesian quadrature tasks

## C.7 Model Sensitivity

To evaluate the sensitivity of our method with regards to the random initialization during training, we run 10 different trials with the same training hyperparameters. The 10 different trained models are evaluated on the regression benchmarks. For each trained model, the same 10 data-splits are used. The average test RMSEs by the different trained models are reported in Fig. 11. For better

visualization, we scaled the median of the RMSEs to 1 for each regression task. In general, AHGP is not very sensitive to the randomness involved in deep learning training. We also observe a slightly bigger variation on dataset with fewer data points.

Figure 11: Violin plot of the test RMSEs evaluated with the 10 trained models. (The violin plot shows the probability density of the RMSEs, smoothed by a kernel density estimator.)