[Reviews · NeurIPS 2020]

Review 1

Summary and Contributions: This paper proposes a hierarchical model over datasets. The model is the form of a transformer which returns the spectral density for the kernel of a GP over the input/output pairs.

Strengths: The combination of ideas is a good one, and the experiments seem well executed. Compared to the related approach of Kim 2019, the method has the advantage that it doesn’t need retraining for new tasks. This is a nice property that could be emphasised more.

Weaknesses: This paper presents fairly a simple combination of existing ideas. The idea of using attention in a meta-learning model for functions was already presented in Kim 2019. When I read the abstract I was hoping that the method might be able to find better parameters than the standard marginal likelihood method. This might happen if the synthetic datasets were generate with a suitable prior for the hyperparameters. I.e. the network is learning how to do non-conjugate bayesian inference rather than just the task to amortizing hyperparameter optimization. This isn’t demonstrated in experiments, so the method only offers and advantage of computation. If computation is the appeal of the method then it needs to be demonstrated in settings where computation is the bottleneck. Unfortunately, to generate synthetic datasets is cubic in the number of points unless approximations are used, so the method would struggle to be applied directly to very large problems. This problem is also shared by neural processes trained on GP samples. The transformer sees each dimension independently, but the function might depend on interactions that are only visible through interactions. E.g. the function f(a, b) = sign(ab) on [-1, 1]^2 would confuse this approach. The regression results are based on RMSE, which is a shame as it neglects comparison of the predictive variance.

Correctness: It all seems correct.

Clarity: The handling of the noise variance wasn’t clear to me. How is noise variance chosen/optimized? I would assume that the transformer would return its value, but that doesn’t seem to be the case. In the appendix it specifies that the training data has fixed noise variance. Why is this done? Surely it would be better to have variable noise or else the transformer will not work well for noise variance significantly different from 0.01.

Relation to Prior Work: Prior work is appropriately referenced. It is a shame that the experiments do not compare to other metalearning approaches e.g. neural processes as this would have been a natural comparison.

Reproducibility: Yes

Additional Feedback: I enjoyed reading this paper and do like the idea. In my view the level of novelty is not quite sufficient for the NeurIPS bar and it has not been demonstrated that the approach adds anything beyond speed of training. As discussed above, if speed is the key advantage then this needs to be demonstrated and the problem of generating appropriate training data discussed. I also have reservations about the way the transformer operates on each dimension independently and might miss interactions between dimensions. It seems to me that this independence is crucial to the dataset generalization property, so I fear it may be a fundamental limitation. Perhaps the authors could clarify this point as I may be missing something. EDIT post rebuttal. I was indeed misunderstanding something, so I have raised my score.


Review 2

Summary and Contributions: This paper introduces a method to amortize the cost of selecting hyperparamters for Gaussian process models. The key insight is that a neural network can find the features that determine good GP hyperparameters from many examples of dataset-hyperparmeter pairs.

Strengths: Selecting the hyperparameters for a GP is a costly exercise, and reducing the cost will be helpful to many GP users. The application to Bayesian Optimisation, where GP hyperparameters need to be optimised often, is particularly compelling. The proposed solution also clearly provides the empirical gains

Weaknesses: I believe that the paper has one main oversight: There is no experiment directly assessing the marginal likelihoods produced by the hyperparmeters produced by the neural networks. Currently there is variation in the performance between many methods which are all effectively the same GP model. In some situations AHGP performs better than direct optimisation. It is important to figure out why. Is it because the neural network is doing a better job at finding a high marginal likelihood? Or is it because it is failing to find the highest marginal likelihood (which is the task it was trained to do), but instead adding implicit regularisation? The authors state that they believe that "this demonstrates the robustness of AHGP when there is not enough data for MLL-opt based approaches to form reasonable point estimates". This can be assessed simply by comparing the marginal likelihoods obtained from AHGP to those from direct optimisation.

Correctness: Yes. Some points: - No consideration is given to selecting an appropriate number of inducing variables. 10% for UCI or 20 for BayesOpt are unlikely to be enough. It is common to keep increasing the number of inducing variables until very little improvement in the ELBO is seen. This gives an unfairly poor representation of the performance of sparse methods, although sometimes a good sparse approximations simply cannot be found. Perhaps a bit of a discussion on this would be good, although I don't expect it to change any of the conclusions. - Matérn kernels do not in general have product kernel structure (as claimed in line 158). E.g. for Matern-3/2, the multiplication of two kernels on different dimensions do not lead to the Matern-3/2 kernel on both dimensions. This is due to the dependence of the kernel on the distance between points, rather than distance squared.

Clarity: Yes, very clear.

Relation to Prior Work: Yes, relations to previous contributions are good. One small point relating to the point on inducing point methods earlier... In addition to the experiments being a bit pessimistic towards inducing point methods, the paper doesn't reference recent work showing that inducing point methods provide 1) arbitrarily good approximations to GP regression models in far less that O(N^3) cost, e.g. O(N polylog N) for the squared exponential kernel 2) without the need for gradient based optimisation of the inducing inputs. The sentence in lines 119-120 "at the cost of a larger and often more challenging optimization problem alongside the potential loss of important information from the dataset" is therefore not quite correct. A small tweak to these sentences and a reference should position the paper correctly to the current state of the literature. As before, this comment does not affect the validity of the method that is introduced.

Reproducibility: Yes

Additional Feedback: This looks like a useful method, and I recommend acceptance. === Rebuttal: Many thanks for checking the marginal likelihood and confirming that it is indeed lower. This confirms that the network fails to optimize the marginal likelihood, while adding a useful regularizing effect.


Review 3

Summary and Contributions: The paper proposes an amortisation scheme to learn hyperparameters for Gaussian process regression. The key argument is that hyperparameter learning is expensive O(N^3) and can find a bad minimum. So if we have access to a bunch of similar datasets to our training set, we can amortise the learning so that it is faster for the training set at hand. The key challenge is to deal with different input cardinalities and dataset sizes, which was dealt with by employing a hierarchical attention network. Some real-world regression and BayesOpt experiments are used to compare the proposed amortised spectral mixture kernel with non-amortised version as well as sparse GPs, in terms of accuracy and run-time.

Strengths: Relevance: The paper considers an important problem for the GP community -- how to efficiently (in time) get a maximum likelihood estimate of the hyperparameters for a GP regression task. I’m not aware of any previous work that “amortises” this estimation procedure with a neural network taking a dataset as inputs. This potentially can be applied to many problem settings, e.g. surrogate modelling for Bayesian optimisation in which we might have obtained “a dataset of datasets” of related functions. Novelty: The use of “hierarchical attention network” with the combination of local transformer, aggregation and global transformer blocks is interesting and is potentially useful for the GP community. But I’m not sure how novel this architecture is to the neural network crowd, or if this has been or could be used for another amortised inference task (e.g. the neural statistician?). Significance: The gain in run-time in practice as shown in the experiment seems significant. It is also interesting that only one inference model was trained on a synthetic meta-dataset and used across different settings, yet, it worked well on average.

Weaknesses: One motivation of the work is that optimising the log marginal likelihood on a single dataset is sensitive to local minima. However, I’m not convinced that the method proposed here could avoid that, as the objective function used to optimise the inference network is also the log-marginal likelihood (and again this function is non-concave) and now that there are a lot more parameters. As a side note, is the local-minimum issue something that cannot be fixed? Existing heuristics for various kernels exist, e.g. for exponentiated quadratic use the median trick, for spectral mixture, use the frequency spectrum of the actual observations and find the peaks. The complexity of the proposed training method seems very high, O(aN^2 + bD^2), i.e. quadratic in each meta-dataset size and input dimension. This is nearly as high as running a single GP regression on each dataset, or more than typical sparse GP on a single dataset. But on the other hand, it could be acceptable if we only train the inference net once on many datasets. That brings me to the point of generalisation to “unseen” dataset. It would be great if this could be more rigorously evaluated. It was not clear from the description in the experiment how the dataset of synthetic datasets were generated. BayesOpt is particularly useful in the case where we only need to query a small number of points and the evaluation of the function of interest is expensive. In this setting, the cost of training the GP is small compared to the cost of one function evaluation. So I’m not sure if the proposed solution here is of practical interest. Recent work on hyperparameter transfer learning and warming up (e.g. Perrone et al, 2018) should be considered as baselines for a fair comparison. Update 1: The authors have sufficiently addressed my concerns. I encourage the authors to include more practical use cases where one might be interested in the proposed method.

Correctness: The claims and method seem correct and sensible to me. Please see above some potential clarification/baselines that could be added to the experiments.

Clarity: The paper is very well written.

Relation to Prior Work: If one key application is Bayesopt, some related work on meta-learning/transfer learning for Bayesopt should be considered/compared to.

Reproducibility: No

Additional Feedback:


Review 4

Summary and Contributions: This paper uses “amortize” inference over kernel hyperparameters in GP regression. They use a single neural network on a set of regression data. This single neural network that is trained on synthetic data is generalized to real-world GP use cases. The neural network provides an estimate of the hyperparameters that usually are calculated with maximizing log marginal likelihood and they call this method AHGP. This work is inspired by “amortized variational inference approach”. However, different from the mentioned paper instead of calculating approximate posterior distribution over the unknown latent variables, they calculate point estimates of the unknown hyperparameters. Kernel wise, they use stationary kernels in the spectral domain and learn the spectral density of the kernel function. In particular they use mixture kernel “Gaussian process kernels for pattern discovery and extrapolation” by Wilson et al. since that is capable of approximating any stationary kernel arbitrarily well. They demonstrate the estimated hyperparameters are comparable with conventional techniques and it is faster. This further accelerates GP regression models and its related applications such as Bayesian optimization and Bayesian quadrature.

Strengths: This paper is technically sound. To the best of my knowledge, the work is novel and relevant to the community.

Weaknesses: As also mentioned by the authors, since they are using deep learning, there is a risk of overfitting and low robustness. This can lead to misspecified hyperparameters and poor prediction performance compared to calculating MLL.

Correctness: The formulation of the model appears to be correct, and I did not spot any particular issues while going through the paper.

Clarity: The paper is well-written.

Relation to Prior Work: This paper clearly discusses how it is different from previous works and how it is related to other works.

Reproducibility: Yes

Additional Feedback:

[Author Response · NeurIPS 2020]

We thank the reviewers for the valuable feedback and address specific comments below.

**Motivation [R3]:** We emphasize that the major advantage of our method is fast identification of reasonable kernel
hyperparameters since MLL-opt is costly and has no global optima guarantees. A side benefit is that our approach
empirically does not result in catastrophically bad hyperparameters because collective training across different tasks
helps avoid bad local minima: averaging the gradient over tasks seems to help the training escape from pathological
solutions. Also, overparameterization in deep neural nets is known to help SGD find good optima both in terms of
optimization and generalization, and sometimes with theoretical support [Allen-Zhu et al., 2019, Du et al., 2019].

**Hierarchical attention architecture [R1 & R3]:** We would like to first clarify that our architecture consists of
three parts: *local transformer*, *aggregation function* and *global transformer*. The *local transformer* sees every
dimension independently and an aggregated representation is calculated for each dimension. The **interactions between**
**dimensions** are later modeled by the *global transformer*. The architecture is carefully designed for the GP problem with
desired properties such as versatility and permutation equivariance. We are not aware of similar existing architecture for
learning on datasets. Kim et al. [2019] uses attention to aggregate information of the context data and relates target
data to context data for prediction. The major difference is that our method uses hierarchical attention and it is able to
take in datasets with **different dimensionality** with **one model**. We will expand this explanation in the paper. We also
expect the proposed architecture to be useful for other tasks that involve learning on sets (e.g., the neural statistician).

**Training setup [R1, R3 & R4]:** With regard to R1's comments on generating large synthetic datasets, we would
like to mention that our model does not necessarily need to be trained on large synthetic datasets for it to generalize
to large-scale problems. In our experiments, we train a single neural model using synthetic datasets generated with
stationary kernels of dimensionality 2∼15 and 30 data points on average. (Full details are available in Appendix C.) This
**single trained model** is applied to **all the real-world tasks** with varying dimensionality and cardinality, yet it is able to
perform fairly well on those **unseen tasks**. This demonstrates the effectiveness of our proposed architecture: although
it is trained on small synthetic datasets, it learns to extract the underlying structure for large unseen datasets. Our model
is expected to work for a relatively low-dimensional GP regression task that follows the stationarity assumption.

**Scalability [R1 & R3]:** Full GP MLL-opt scales as $\mathcal{O}(N^3)$ using Cholesky decomposition (GPy) and $\mathcal{O}(TN^2)$ using
the conjugate gradient method (GPyTorch). It can be reduced to $\mathcal{O}(NM^2)$ with a sparse GP. All the methods require
iterative optimization, meaning the computation cost incurred in practice is the complexity multiplied by the number of
opt iterations. In comparison, our method scales as $\mathcal{O}(N^2 + D^2)$ and only requires one forward pass of the trained
neural net. In terms of actual running time, our method compares much more favorably with all the baselines above.
The $2,000$ datapoints used in our experiments are already considered large-scale for GP. To give a larger-scale example,
for *power plant* with $4,000$ data points, our method is 226 times faster than GPyTorch on GPU while 332 times faster
than GPy on CPU. We will include these larger-scale experiments in the revised version. It is possible to go even larger
for our model, but full GP MLL-opt methods either take too long on CPU or report out of memory issues on GPU.
Note that our model's complexity can be further reduced to $\mathcal{O}(MN + MD)$ if $M$-sparse attention is introduced. With
regards to R3's comments that BO is usually small-scale, it might be observed that 1) BO is small-scale in part because
of the cost of hyperparameter inference, and 2) AHGP is often empirically more robust in the small-data regime.

**Predictive variance [R1]:** It was presented in Appendix C. We will improve the presentation to make it more accessible.

**MLL when AHGP performs better than MLL-opt [R2]:** Upon checking, we observed that AHGP found an MLL
that is slightly smaller than that of MLL-opt. This illustrates that MLL-opt can potentially overfit in the low-data regime,
while AHGP adds extra regularization through learning to optimize over many different GP tasks.

**Noise variance [R1]:** In all experiments, we normalize the data input and output. The noise variance is fixed at 0.01. It
is possible for our method to learn the noise variance. In practice, we find the predictive performance is not sensitive to
the noise variance and setting it to 0.01 gives competitive performance for all baselines. Also, spectral mixture kernel is
flexible enough to model the noise as well (with $\mu = 0$, small $\sigma^2$). We will add the clarification to make it more clear.

**Neural processes (NPs) [R1]:** In terms of the problem formulation, our work targets efficient estimation of GP
hyperparameters as part of a full GP inference procedure, while NP serves as an end-to-end approximation to the GP
inference itself. We also hope to emphasize that our method tackles a richer distribution of tasks (i.e., regression tasks
generated from stationary GP, with **different dimensionality** and cardinality) with **a single trained model**, while NP
aims at solving one particular task or similar tasks (**same dimensionality**) with one neural model.

**BO hyperparameter transfer learning [R3]:** Thanks for the pointer. We will include them in related work. This type
of approach focuses on better warm-starting BO through leveraging data of previous related BO runs, such that a good
surrogate model can be built with very few evaluations of the target function. In comparison, our method focuses on
fast inference of kernel hyperparameters, and our single trained model is applicable to a wider range of GP use cases.

**Inducing point methods [R2]:** Thanks for the pointer. We will include the reference on efficient inducing point
method [Burt et al., 2019] and add experiments with an increasing number of inducing points for sparse GP.

[Meta-Review · NeurIPS 2020]

All four reviewers were in favor of accepting this paper. Despite the fact that this paper is not very dense in terms of methodological contributions, the reviewers appreciated the simplicity of the idea, which also gives good results. Indeed, amortization for the hyperparameters of the GP has not been considered before, and seems like a neat solution to the key problem of learning GP models. Furthermore, the reviewers appreciate the potential applicability of this method, e.g. to Bayesian Optimization.